# Frequency and Uncertainty driven Deep Learning Approach to Segment Electrocardiogram Signals for Effective Heart Parameters Estimation

Sayna Rotbei
*Department of Electrical Engineering and Information Technology*
*University of Naples, Federico II*
Naples, Italy
sayna.rotbei@unina.it

Gennaro Esposito Mocerino
*Department of Electrical Engineering and Information Technology*
*University of Naples, Federico II*
Naples, Italy
gennaro.espositomocerino@unina.it

Muhammad Salman Haleem
*School of Electronic Engineering and Computer Science*
*Queen Mary University of London*
London, UK
m.haleem@qmul.ac.uk

Leandro Pecchia
*School of Engineering*
*University Campus Bio-Medic*
Rome, Italy
leandro.pecchia@unicampus.it

Alessio Botta
*Department of Electrical Engineering and Information Technology*
*University of Naples, Federico II*
Naples, Italy
a.botta@unina.it

*Abstract*—Accurate classification of electrocardiogram signals is reliant on accurate heart rhythm parameters detection, which requires effective electrocardiogram segmentation at the beat level. In this study, we propose and evaluate integrating temporal, frequency, and uncertainty-informed deep learning approaches for classifying electrocardiogram signals into three main categories: PQ, QRST, and TP. Utilizing PhysioNet's QT public database, we preprocess the data, including noise filtering, gap removal, and normalization, to prepare it for deep learning model input. We employ multilayer deep learning architectures, integrate them with additional features such as short-time fourier transform and approximate entropy to determine uncertainty-enhanced classification performance. Through extensive experimentation and five-fold cross-validation, we analyze the impact of layer duplication and additional features on classification accuracy. Our results demonstrate that our proposed methodologies achieve up to 96% accuracy in classifying Electrocardiogram (ECG) signals, providing valuable insights for improving heart rhythm parameter detection and suggesting avenues for further research in ECG signal processing.

*Index Terms*—Artificial intelligence, Deep Learning, Electrocardiogram Signals, Heart Rhythm,

## I. Introduction

Cardiovascular diseases are responsible for a significant number of deaths globally [1]. Cardiac arrhythmia is one of the most common causes of cardiovascular disease, in which the heart beats in a manner that differs from standard patterns [2]. The most common method of diagnosing and following up on patients is the analysis of critical portions of their electrocardiogram (ECG)s [3]. An ECG signal records the electrical activity of the heart with the help of electrodes placed on the body [4]. It is a non-invasive device used to record the direction and magnitude of the electrical activity of the heart that occurs as a result of the depolarization and

repolarization of the atria and ventricles [5]. The variations in heart characteristics directly impact the ECG, making it a highly unique biometric system. Therefore, it is crucial to consider the differences when analyzing ECG signals for diagnostic purposes or biometric identification [5,6].

The accuracy of segmenting the ECG beats and detecting fiducial points are pivotal for analyzing ECG signals and knowing certain cardiac conditions [3]. Clinical practitioners currently analyze the ECG signals manually to determine these cardiac conditions [7]. It is, however, a time-consuming and skill-intensive process to interpret data manually [2]. Incorrect interpretation of ECG signals may result in incorrect clinical decisions and may endanger the life and health of patients. Identifying the onsets and offsets of QRS complexes, as well as P and T waves in an ECG signal, is known as segmentation or delineation [4]. To address these, research is being conducted to automate the ECG delineation process.

There are certain challenges. Firstly, personalized ECG signal heterogeneity questions the robustness of state-of-the-art method. Secondly, most of the existing models only accommodate temporal variations [3] [4]. Most of the signal variability may be reflected in the frequency domain, which can be considered for improved segmentation performance. Thirdly, ECG signal with different chronic conditions may experience different beat parameter variation [8].

This study aims to improve ECG segmentation performance using a Deep Learning (DL) model that considers frequency spectrum and uncertainty. The model calculates the ECG frequency spectrum and beat entropy, and then applies a Convolutional Bidirectional Long Short Term Memory (Conv-BiLSTM) model using this information. This paper is the first to automatically compute frequency spectrum and

entropy features to enhance results and reduce computation time. The goal is to create a reliable DL method that needs less computation time and is robust to noise.

The paper is organized into six sections. Following the introduction, the article is structured as follows: in the section II, the paper critically examines the recent literature related to ECG signal analysis, providing an in-depth analysis and comparison of various studies. Moving on to section III, the dataset and the detailed methodology used to obtain results are meticulously discussed, including the specific parameters and techniques employed. Subsequently, section IV delves into a comprehensive discussion of the obtained results, highlighting key findings and their implications. Finally, sections V and VI encompass a thorough discussion and conclusion, offering comprehensive insights and implications derived from the study.

## II. LITERATURE REVIEW

As a fundamental diagnostic tool in clinical practice, the ECG provides a wealth of information through the analysis of its signals over time. [9]. Efficiently extracting and interpreting this information hinges on the precision of signal segmentation, a process crucial for delineating distinct cardiac events such as the P, QRS, and T waves [10]. This literature review critically examines the evolving landscape of ECG signal segmentation techniques into contemporary methodologies by providing an overview, shedding light on the diverse approaches and innovations that have shaped the current state of ECG signal segmentation.

Initially, different signal processing techniques based on differentiation, transformation, and derivation methods have been studied. While some of these techniques might excel at detecting QRS complex in ECG signals, they might not be as versatile or widely applicable to other tasks or aspects of ECG analysis [11]. One of the main drawbacks of Machine Learning (ML)-based methods is their reliance on a set of meaningful features, which can be time-consuming and sometimes impossible to obtain in real-world scenarios [11]. One of the approaches implemented a simple Linear Regression (LR) process and analyzed 260 ECG signals, achieving an average sensitivity for identifying all peaks. To validate the robustness of the approach, an ECG sensor was developed to acquire real-time signals, visually represented based on the implementation of the detection algorithm. Despite a high average sensitivity, there was a 3% error rate in the correct detection of the Q point. The dataset used for training and testing had limited examples of non-sinusal or abnormal beats, suggesting a need for more diverse ECG data to enhance the algorithm's generalization capacity. They mentioned in their paper that random forest was the best ML method [12].

Recently, deep learning (DL) based methods have been widely used in recent years to overcome the mentioned challenges and classify ECG signals according to their waves [11]. In the study conducted by Jikuo Wang et al., [13] a new Convolutional Neural Network (CNN) with a non-local convolutional block attention module has been proposed to classify ECG heartbeats automatically with high accuracy. Hedayat Abrishami et al. [14] proposed a method for segmenting ECG signals, a critical task in cardiology and pharmaceutical studies for predicting heart symptoms and medication effects. Although DL methods have effectively classified heart conditions, there is a dearth of DL-based approaches for characterizing ECG temporal features. The suggested method utilizes a Recurrent Neural Network (RNN) with Long Short-Term Memory (LSTM) layers, categorizing each ECG sample into P-wave, QRS-wave, T-wave, or neutral (others). The research demonstrates that DL sequence learning methods, particularly employing simple local features, outperform traditional Markov models in terms of accuracy. Remarkably, in T-wave segmentation, the approach achieves 90% accuracy, compared to the 74.2% accuracy attained by Markov models.

Apart from analyzing time-series signals in the time domain, there have been several attempts to analyze time signals in frequency and power spectrum domains. For example, one of the previous attempts suggest the cardiac arrhythmias classification using the two-dimensional convolutional neural networks (2D-CNNs) with time-frequency spectrograms of ECG waves obtained through Short-Time Fourier Transform (STFT). This method eliminates the need for manually extracting features from ECG recordings. For ECG wave detection, Beraza and her colleague [3] utilized discrete wavelet transform for detecting QRS complex onset and offset. Their another algorithm utilized multi-scale morphological derivate demonstrated high sensitivities for detecting P-wave peak, onset, and offset detection, respectively. Besides, they utilized the Phasor transform for T-wave segmentation with high segmentation accuracy. Gupta and Mittal [15] propose an innovative method for automated QRS complex detection in ECG recordings. Their integrated approach combines chaos analysis, STFT, and Principal Component Analysis (PCA), achieving high accuracy in the analysis of data from the PhysioNet database. This method shows potential for early diagnosis of cardiac diseases and optimization of patient monitoring and treatment processes.

A couple of studies utilized entropy to detect ECG beat segments. Malali et al. [16] proposed a neural model based on Convolutional Long Short-Term Memory (ConvLSTM) for ECG waveform segmentation. Trained on the PhysioNet QT dataset, the model outperformed traditional methods such as Hidden Markov Model (HMM). The model architecture includes a convolutional layer followed by a bidirectional LSTM layer, with the addition of a self-attention layer to enhance performance. Results indicate that the ConvLSTM model surpassed other ECG segmentation models, demonstrating its effectiveness in automated ECG analysis and cardiac diagnoses. Aboli N. et al. [11] has introduced a new approach for segmenting ECG waves, termed semantic segmentation utilizing entropy, commonly employed in image segmentation. The average and weighted accuracies surpassed the performance of LSTM, Bidirectional Long Short-Term Memory (BiLSTM), and double BiLSTM with increments in a weighted average accuracy of 12.28%, 8.08%, and 5.77%,

respectively.

Despite significant advancements in traditional and ML/DL approaches, many methods continue to depend on manual feature extraction or require substantial computational resources. In this study, we introduce a novel approach that leverages the strengths of STFT for feature extraction, integrated with deep learning models, specifically incorporating approximate entropy as a feature for ECG signal segmentation. Our method aims to enhance robustness by employing STFT and approximate entropy to improve segmentation resilience to noise; reduce computation time by minimizing dependence on extensive computational resources and manual intervention, addressing a common limitation of existing methods; and achieve high accuracy by maintaining or surpassing the performance of current state-of-the-art methods, while offering a more practical and efficient solution for real-time applications. In conclusion, this work advances the field by addressing the key limitations of existing methods and provides a balanced approach that combines efficiency with robustness.

## III. Materials and Methodology

We provided the DL models with 3-dimensional arrays as input data. Additionally, we incorporated the use of STFT and approximate entropy as supplementary features to enhance the performance of the model during the signal analysis process.

### A. Database

The PhysioNet's QT database [17] was utilized to evaluate proposed algorithms.

This dataset includes 105 ECG recordings selected from various databases, showcasing a diverse range of QRS, T-wave, and P-wave morphologies. In each record, at least 30 beats were manually annotated by an expert. Furthermore, 11 of the recordings were annotated by a second expert (It should be mentioned that only one of the annotations was used in this study) . This database contains annotations on the peak, onset, and offset of the QRS complex (QRSpeak, QRSon, QRSoff) and the P-wave (Ppeak, Pon, Poff), as well as the peak and offset of the T-wave (Tpeak, Toff). Only some records (totaling 1345 beats) contain the onset of the T-wave (Ton). A total of 94 records were used, and 11 records were excluded during the pre-possessing process due to annotation issues; some of the annotations were not available for some of these records.

In terms of the quality of the dataset, due to the unclear criteria followed by physicians in locating fiducial points and the lack of consensus in the scientific community regarding the precise boundaries of a P-wave, it is often not feasible to definitively classify an annotation as erroneous or imprecise. Nonetheless, there are certain specific cases where it is indisputable that the annotations are severely defective [18].

### B. Pre-processing

As an initial step in this research endeavor, the recorded files and their corresponding annotations were examined, followed by a visualization of the signal plots to facilitate a deeper understanding of the dataset. This allowed for a more comprehensive evaluation of the information and paved the way for further analysis. In the subsequent phase of the experiment, the signals were subjected to a filtering process utilizing two techniques, namely wavelet decomposition and denoising.

*1) Wavelet Decomposition and Denoising:* The wavelet transform is a technique used to represent a signal in a two-dimensional function of time and scale [19]. It achieves this by using the correlation with the translation and dilation of a wavelet function. The wavelet transform represents a signal as a sum of wavelets with different locations and scales. This allows for the use of long time intervals for low-frequency information and shorter regions for high-frequency information. The reconstruction process can be performed after discarding undesired wavelet coefficients that contain random noise [19].

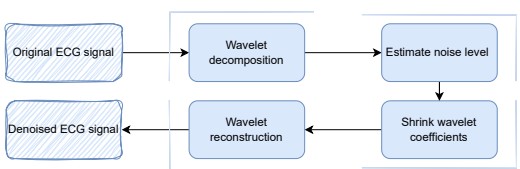

Fig. 1. Wavelet denoising procedure.

Fig 1 illustrates the basic idea of a wavelet-based denoising procedure. The process involves three main steps: decomposition, modification of coefficients, and reconstruction. This approach was implemented to enhance the quality of the signals and to minimize the presence of noise. The wavelet decomposition process involves breaking down the signals into various sub-bands, each representing different frequency ranges, and filtering out unwanted frequencies. This approach was adopted to improve the accuracy and reliability of the results obtained from the experiment. Once the extraneous noise had been removed from the signals, the segments were divided into portions of equal size. Fig 2 compares one of the original ECG signals with the pre-processed one.

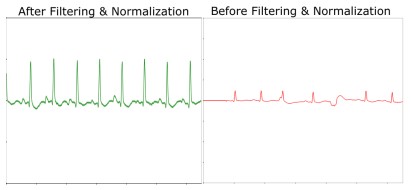

Fig. 2. Effect of preprocessing, comparing original ECG signal with the signal after pre-processing

*2) Signal Normalization:* In the course of preprocessing, the subsequent measure entailed normalizing the complete dataset by the z-score normalization method, which is the critical step ensuring that the dataset variables are not biased by their original scale or measurement unit. Therefore, the normalized dataset provides a more accurate and reliable basis for subsequent analysis and modeling. There are dozens of

normalization methods in the literature. The optimal approach is not universally applicable and is contingent upon the specific characteristics of the problem and the intrinsic attributes of its data. Prevalent literature suggests that z-normalization is universally suitable across domains, problems, algorithms, and data [20]. The z-score normalization is required in deep learning approaches as the output score lies in the range of -1 and 1.

*3) Balanced Dataset Creation:* To address the data imbalance problem, we performed a series of merge operations. Specifically, we merged the "P-wave" and "PQ-wave" to form a single entity. Furthermore, we combined the "QR-wave", "RS-wave", and "ST-wave" into a unified entity. Finally, we merged the "T-wave" and "TP-wave" to create a cohesive entity, which resulted in a more balanced dataset with significantly improved accuracy. Table I includes a summary of

TABLE I
DISTRIBUTION OF INPUT

| P, PQ | QR, RS, ST | T, TP |
|-------|------------|--------|
| PQ | QRST | TP |
| 89986 | 80982 | 148832 |

the methods used to create a balanced dataset, along with the distribution of each class. This information is vital in gaining a deeper understanding of the composition of the dataset and can be used to inform future analyses. It is imperative to consider the distribution of each class when working with imbalanced data, as it can significantly impact the accuracy of predictive models.

In this study, the 5-fold cross-validation method was employed to enhance the stability of estimates and to achieve more readily generalizable model performance. The dataset was partitioned into 5 equally sized subsets, denoted as folds. One subset was exclusively allocated for testing the model's performance, while the remaining 4 subsets were utilized for training the model. This iterative process was replicated 5 times, wherein the training and test sets were varied for each fold. Ultimately, each of the 5 subsets served once as an exclusive test set for evaluating the performance of the model.

### C. Model Development

Several methods have been developed in order to achieve better results for classifying ECG signals in three different classes mentioned in Table I. The datasets were inherently complex and fell within the category of substantial volumes of data. The primary objective was to achieve optimal outcomes, necessitating the utilization of sophisticated mathematical computations inherent to DL methodologies. Moreover, DL methods work better for sequential data learning (like ECG) where data points come in a sequence. This is due to their power of learning local features based on temporal context. Fig 3 depicts the DL architecture used in this study. Also, through a comprehensive review of the existing literature, we

conducted an in-depth exploration of various features, encompassing diverse forms of entropy, standard deviation, mean values, and others. Notably, our investigation revealed that leveraging approximate entropy and STFT produced superior outcomes amidst the array of features considered. [21][22]. The details are as follows:

*1) Entropy:* Entropy measures serve as a valuable tool to gain insight into the intricacy or regularity of ECG signals. These measures can be used to quantify the degree of disorder or complexity of ECG signals, providing a meaningful metric for evaluating their quality. The development of this measure was aimed at assessing the complexity or randomness inherent in datasets, especially in the context of physiological signals such as ECG. In this approach, for each window of 100 elements $\{x_i, x_{i+1}, \ldots, x_{i+99}\}$, the approximate entropy $\text{ApEn}(100, r, 100)$ is computed and stored in the entropies list. This is done iteratively across the entire sequence, resulting in a list of approximate entropy values that reflect the complexity and regularity of the sequence over time. Mathematically, assume the ECG signal sequence as $[x_1, x_2, \ldots, x_N]$, we define a vector of length $q$ $[u_i = [x_i, x_{i+1}, \ldots, x_{i+q-1}]$ for $i = 1, 2, \ldots, N - q + 1$. The approximate entropy can be given as

$$\text{ApEn}(q, r, N) = \Phi^q(r) - \Phi^{q+1}(r)$$

. Where $\Phi^q(r) = \frac{1}{N-q+1} \sum_{i=1}^{N-q+1} \ln(C_i^q(r))$ is the average similarity with $C_i^q(r) = \frac{\text{number of } j \text{ such that } d(u_i, u_j) \leq r}{N-q+1}$ as similarity measure, $r$ is a tolerance value, and distance between vectors $u_i$ and $u_j$ is represented as $d(u_i, u_j) = \max_{k=0,\ldots,q-1} |x_{i+k} - x_{j+k}|$.

*2) Short Time Fourier Transform:* STFT is the technique employed to analyze the frequency content of a signal over time. The STFT is widely used in a variety of fields, including audio and speech processing, image processing, and biomedical signal analysis. In analyzing the ECG signal, the STFT can be used to examine the frequency components present in an ECG signal at different time intervals.

In this procedure, the signal is extended to ensure its length is a multiple of the frame size. The extent of the extension is calculated as the difference between the frame size and the remainder of the signal length divided by the frame size. This padded signal is generated by adding zeros to the end of the original signal. Then, the padded signal undergoes a reshaping process into a 2D array, wherein each row accurately represents a frame of the signal. Finally, the STFT is computed by performing the Fast Fourier Transform (FFT) for each frame along the rows (axis=1) and subsequently obtaining the magnitude of the FFT results. In terms of mathematics, the STFT is represented as follows, where: $x[n]$ is the input signal, $N$ is the frame size (length of each frame), $m$ is the frame index, $k$ is the frequency bin index, $H$ is the hop size (the number of samples between successive frames), which in the provided function is implicitly equal to the frame size since

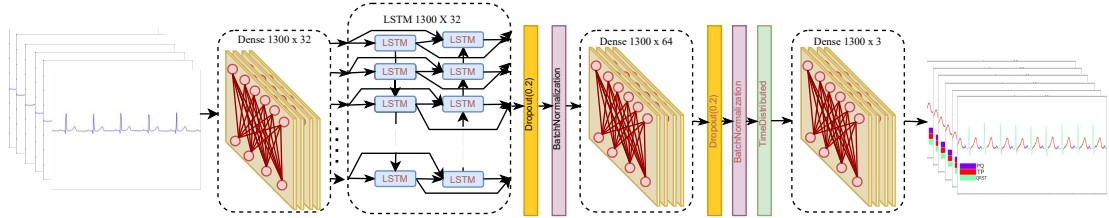

Fig. 3. The architecture used for segmenting ECG signal, the first experiment

no overlapping is mentioned, and $j$ is the imaginary unit.

$$\text{STFT}(m,k) = \left| \sum_{n=0}^{N-1} x[n+mH] \cdot e^{-j\frac{2\pi kn}{N}} \right|$$

In summary, the process of SSTFT entails padding the signal to fit an integer number of frames, reshaping the signal into frames, and subsequently applying the FFT to each frame in order to calculate the magnitude of the frequency components.

*3) Recurrent Neural Networks:* Our approach entailed the utilization of Recurrent Neural Networks (RNNs) to segment the entire ECG beat to facilitate the modeling of the event sequence. This approach was chosen due to the inherent ability of RNNs to develop a deep understanding of sequential data by processing the inputs [23]. By adopting this approach, we were able to achieve a high degree of accuracy in the segmentation of ECG beats, enabling more precise analysis and interpretation of ECG data.

In response to the long-term dependency issue in traditional RNNs, among these architectures, LSTM was developed to overcome the problem [23]. The LSTM architecture allows information to flow through the network over extended periods, thereby avoiding the vanishing gradient problem observed in conventional RNNs.

*4) Convolutional Neural Network:* We have developed a Conv-BiLSTM architecture to extract local and time-variant features. This architecture is trained on data segments that capture information from past and future contexts, enabling the model to learn complex temporal patterns and improve its performance in capturing the data dynamics. Our approach involves applying a one-dimensional convolutional layer that convolves the signal $x_{1d}$ with the kernel initialized by the 'glorot-uniform' method in the temporal dimension. The kernel weights $W_{1d}$ are updated after each iteration to ensure optimal performance. These can be represented as:

$$y_{1d} = [W_{1d} \otimes x_{1d} + b]$$

Moreover, implementing a Conv-BiLSTM network has been undertaken to facilitate the training of temporal context in both directions of time. This approach enables the signal to be trained based on the temporal context, thereby enhancing the accuracy and effectiveness of the training process. Moreover, it enables the network to better capture the long-term dependencies and temporal correlations within the data. This can be expressed as follows (It should be mentioned that $\vec{h}_t$ and $\overleftarrow{h}_t$ are forward and backward LSTM outputs at time $t$):

$$h_t = Dropout\left(\left[LSTM\left(y_{1d_t}, \vec{h}_{t-1}\right), LSTM\left(y_{1d_t}, \overleftarrow{h}_{t+1}\right)\right]\right)$$

*5) Attention Block:* Additionally, we used the self-attention block. This study describes a block aimed at temporally correlating ECG sequences to compute abstract representations as a function of bidirectional temporal context. Correlations between the ECG sequences can be used to represent attention weights. A bidirectional temporal context of ECG sequences can be leveraged in this method for extracting meaningful information and enhancing the accuracy of ECG-based applications. It should be clear that dot $(h_{t_i}, h_{t_j})$ is the attention score, which is the dot product of LSTM output by itself; J is the total number of LSTM outputs, $\sigma_{max}$ the softmax function representing the attention weights. Weights for attention can be calculated as follows:

$$\sigma_{max}(z)_i = \frac{e^{dot\left(h_{t_i}, h_{t_i}\right)}}{\sum_{j=1}^{J} e^{dot\left(h_{t_j}, h_{t_i}\right)}}$$

Also, the dense layer with the ReLU activation method was used to introduce the feature of nonlinearity to a DL model and resolve the issue of vanishing gradients.

*6) Experimental Settings:* In order to assess the efficacy of the adopted technique in optimizing the performance of the model, we increased the number of layers by duplicating the same layers and changing some parameters. This was done to determine the extent to which the method can be employed to enhance the overall performance of the model. For the primary try, we defined a dense neural network layer that had 32 neurons, and we applied L2 regularization to the weights of the layer. After that layer, we decided to consider an BiLSTM with 32 units that are capable of generating output sequences based on the input sequences provided. In order to prevent overfitting, 20% of the outputs of neurons randomly was set to zero. Then, we considered improving training speed, stability, and generalization performance by adding a "BatchNormalization" layer. To improve the generalization ability, Non-linearity, and control overfitting of the model, we defined a dense layer with 64 neurons, used ReLU activation and applied L2 regularization to the weights with a regularization strength of 0.001. The forthcoming three layers were constructed by employing the last three layers. Then, in order to classify output based on probabilities, we used a dense output layer with dimout neurons and a softmax activation function. Finally, the model complied with the Adam optimizer by using categorical cross-entropy as the loss function and accuracy as the evaluation metric during training. Through experimentation and empirical analysis, we had three dense neural network layers, one with 32 and the others with 64 neurons, and 'ReLU' as the activation method. Also, we had three BiLSTM layers with the same configuration. Further, two layers were added to protect against overfitting by setting 20% of outputs to zero.

*7) Performance Evaluation:* In order to evaluate the performance of the employed methods, a five-fold cross-validation procedure was implemented. A confusion matrix was utilized to obtain the resulting evaluation metrics. This approach allowed for a rigorous assessment of the predictive power of the model and facilitated the identification of any potential weaknesses in the methodology. The use of cross-validation and confusion matrices is a standard practice in the field of Artificial Intelligence (AI) and ensures that the results

are reliable and reproducible. Fig 4 illustrates the methods used to achieve the results, along with what was utilized to achieve those results. The present study aimed to enhance the performance of DL method in the classification of ECG signals. To achieve this, four different experiments were conducted. The ensuing section presents the obtained results. The results of the experiments provide valuable insights into the efficacy of various approaches in improving the accuracy of the DL methods for ECG signal classification. A technique utilizing a random seed was employed to ensure consistency between the training and test sets. The purpose of this technique was to maintain the integrity of the data and reduce the probability of errors or bias in the results. By implementing this approach, we were able to establish a standardized process for selecting and dividing the data into respective sets. This methodology has been widely accepted within the academic and business communities as a reliable means of achieving accuracy and consistency in data analysis.

## IV. Result

We performed the experiments based on a combination of DL layers and features. Table II shows the result of the state-of-the-art along with the obtained results of our architectures for all five-fold cross-validation. Upon duplication and increasing the size of the model, the performance of the developed method exhibited some improvement. This result suggests that model scaling may be a viable technique for enhancing the performance of the model. The results also indicate that increasing the size of the model, despite the additional computation requirements, leads to a higher degree of performance. Notably, the analysis reveals that the highest performance levels achieved for the 'PQ', 'QRST', and 'TP' classes were approximately 96.4%, 94.3%, and 96.6%, respectively. Moreover, the model was able to attain an average accuracy level of up to 96%. The outcomes of this part provide insights about the impact of the model size on its overall performance.

In order to understand better the behavior of the used models, Table III shows the confusion matrix of the best 4 experiments. Also, we present the results in the form of other metrics such as precision, recall, and F1 score in Table IV. The results indicate that the involvement of frequency patterns and measures of uncertainty can improve the segmentation performance, especially in the case of the QRST complex. This is due to the reason that the ST segment has little temporal information to be identified and thus requires frequency and uncertainty-based information for improved segmentation performance. This can also be visualized in Figure 5, which shows involving the entropy and frequency spectrum results in segmentation performance closer to the ground truth.

Upon analysis, it is evident that the integration of STFT into the database as a feature resulted in an improvement in accuracy compared to solely incorporating entropy within the dataset. Therefore, in terms of feature significance, STFT surpasses entropy. Nevertheless, the combined utilization of these two features demonstrated optimal results.

Finally, we tested our architecture to assess the accuracy and precision across different heart conditions and ECG recording types. Note that the QT Database is composed of different types of ECG recordings collected under normal and different heart conditions. The results have been presented in Table V and Table VI. The results show high segmentation performance in most of ECG recording types. However, in the case of Supraventricular Arrhythmia, the segmentation performance was dropped due to the influence of different frequency components, which were not detected by entropy measures or STFT.

## V. Discussion

Accurately detecting critical parameters of ECG signals is of utmost importance as the consequences of any errors could potentially jeopardize the life and well-being of the patients. Any inaccuracies or errors in ECG signal detection can have serious implications and may lead to adverse health outcomes for patients. Moreover,

### TABLE II
State-of-the-art along with mean and standard deviation of the performances across our different architectures

| Architectures | PQ | | QRST | | TP | |
|---|---|---|---|---|---|---|
| HMM on raw ECG data [24] | 5.5 | | 79 | | 56.03 | |
| HMM on wavelet encoded ECG [25] | 74.20 | | 94.40 | | 88.23 | |
| BiLSTM [26] | 93.83 | | 95.74 | | 90.30 | |
| Conv-BiLSTM [27] | 94.69 | | 96.04 | | 91.78 | |
| Conv-BiLSTM-Attention [27] | 94.77 | | 97.05 | | 91.77 | |
| EXP_1: ECG + multilayer LSTM | 96.11 | ± 1.0 | 94.136 | ± 1.60 | 96.792 | ± 0.85 |
| EXP_2: ECG + entropy + multilayer LSTM | 95.756 | ± 1.02 | 92.494 | ± 1.43 | 96.47 | ± 1.07 |
| EXP_3: ECG + STFT + multilayer LSTM | 96.786 | ± 0.63 | 93.55 | ± 1.36 | 96.29 | ± 1.0 |
| **EXP_4: ECG + entropy + STFT + multilayer LSTM** | **96.4** | **± 0.52** | **94.308** | **± 1.36** | **96.584** | **± 1.2** |

### TABLE III
Confusion matrix of the top 4 experiments

| *Actual\Predicted* | | PQ | TP | QRST |
|---|---|---|---|---|
| | PQ | 85875 | 2018 | 1466 |
| EXP_1 | TP | 1571 | 142921 | 3163 |
| | QRST | 1970 | 2787 | 75429 |
| | PQ | 85557 | 1542 | 2260 |
| EXP_2 | TP | 1786 | 142469 | 3400 |
| | QRST | 1677 | 4368 | 74141 |
| | PQ | 86490 | 1516 | 1353 |
| EXP_3 | TP | 2091 | 142161 | 3403 |
| | QRST | 1923 | 3274 | 74989 |
| | PQ | 86142 | 1135 | 2082 |
| EXP_4 | TP | 1569 | 142598 | 3488 |
| | QRST | 1687 | 2900 | 75599 |

accurately detecting ECG parameters is a crucial initial step in predicting patients' future health issues. By correctly identifying the ECG parameters, healthcare professionals can determine the current cardiac condition of patients and predict potential health issues that may arise in the future. This information can help healthcare providers develop individualized treatment plans for patients, as well as provide early intervention and preventive care to improve patient outcomes. Therefore, it is important that healthcare providers have the necessary tools and expertise to accurately detect ECG parameters and interpret the results to provide the best possible care for their patients. Therefore, it is imperative to establish a robust and dependable classification method for classifying the parameters of ECG signals with a high accuracy and low error rate.

In this paper, we classified the ECG signal in order to detect the heart rhythm parameters. We worked on classifying the ECG signal into three main classifications, including 'PQ', 'QRST', and 'TP', through four different methodologies. We applied all proposed methodologies to PhysioNet's QT public database to test our models. In the first step, we preprocessed the mentioned public dataset by filtering the noises, removing probable gaps in signals, and normalizing them. Then, we feed the DL models with the 3-dimensional arrays. Moreover by analyzing the signal at hand STFT and approximate entropy were used as the additional variables for improving the performance of the model. By employing the suggested methods, the performance of the classification task improved to 96%. This shows that involving the frequency-based information and uncertainty along with the temporal information improves the time-series signal analysis and classification. Table IV outlines the metrics used to evaluate the performance of the top 4 models and the achieved results,

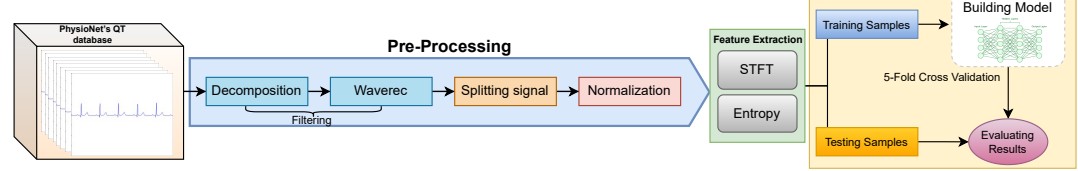

Fig. 4.   Utilized items in the study

and Fig 5 the achieved improvement through the implementation of methodologies. It should be mentioned that all results were generated using a random seed; thus, they are generalizable.

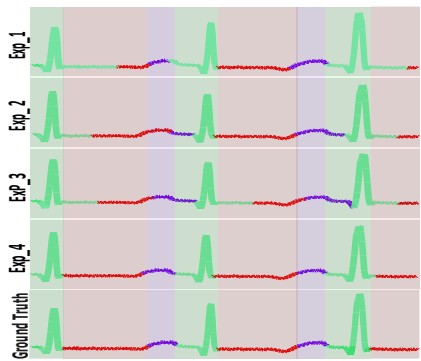

Fig. 5.   Visualization of the improvement in various used architectures

TABLE IV
THE PERFORMANCE OF THE 4 BEST EXPERIMENTS

|  | Metric | PQ | TP | QRST | Accuracy overall |
|---|---|---|---|---|---|
| **EXP_1:** ECG+ Multilayer LSTM | Precision | 0.960 | 0.967 | 0.942 | 0.9591 |
|  | Recall | 0.961 | 0.968 | 0.941 |  |
|  | F1 Score | 0.960 | 0.967 | 0.941 |  |
| **EXP_2:** ECG + Entropy + Multilayer LSTM | Precision | 0.961 | 0.960 | 0.929 | 0.9526 |
|  | Recall | 0.957 | 0.965 | 0.925 |  |
|  | F1 Score | 0.959 | 0.962 | 0.927 |  |
| **EXP_3:** ECG + STFT + Multilayer LSTM | Precision | 0.956 | 0.967 | 0.940 | 0.9572 |
|  | Recall | 0.968 | 0.963 | 0.935 |  |
|  | F1 Score | 0.962 | 0.965 | 0.937 |  |
| **EXP_4:** ECG + STFT + Entropy + Multilayer LSTM | Precision | 0.964 | 0.972 | 0.931 | 0.9595 |
|  | Recall | 0.964 | 0.966 | 0.943 |  |
|  | F1 Score | 0.964 | 0.969 | 0.937 |  |

Furthermore, the importance of analyzing signals ECG and adding other parameters to the current public dataset was explored, demonstrating that heart rhythm parameters can be classified accurately, achieving a high level of reliability and a low error rate.

These findings could have significant implications for the design and development of more precise and reliable ECG-based diagnosis systems. Notwithstanding the classification potential of the described methodologies and the opportunity to unveil the role of duplicating DL model layers for analyzing ECG signals and introducing novel parameters to the dataset, it is imperative to be cautious in light of certain considerations. Although the present study aims to analyze the classification performance of signals ECG, the results of this study are based solely on a single public dataset and, therefore, may not accurately reflect a larger population or larger dataset.

TABLE V
ACHIEVED ACCURACIES ACROSS DIFFERENT HEART CONDITIONS AND
ECG RECORDING TYPES ON OUR ARCHITECTURE

| Databases from different heart conditions and recording types | Total Records | PQ (%) | QRST (%) | TP(%) |
|---|---|---|---|---|
| European ST_T | 33 | 95.40 | 93.65 | 98.10 |
| MIT-BIH Arrhythmia | 15 | 98.18 | 99.04 | 97.30 |
| MIT-BIH Long-Term ECG | 4 | 98.17 | 97.08 | 98.30 |
| MIT-BIH Normal Sinus Rhythm | 10 | 98.58 | 96.52 | 98.69 |
| MIT-BIH ST Change | 6 | 96.90 | 96.12 | 98.02 |
| MIT-BIH Supraventricular Arrhythmia | 13 | 95.04 | 69.50 | 88.89 |
| Sudden Death Patients BIH | 24 | 97.63 | 97.42 | 99.07 |

TABLE VI
THE PERFORMANCE OF OUR OPTIMAL ARCHITECTURE ACROSS
DIFFERENT HEART CONDITION AND ECG RECORDING TYPES VIA OTHER
PERFORMANCE METRICS

| Heart Conditions and ECG recording types | Precision | F1_Score | Recall |
|---|---|---|---|
| European ST_T | 0.9607 | 0.9589 | 0.9571 |
| MIT-BIH Arrhythmia | 0.9812 | 0.9814 | 0.9814 |
| MIT-BIH Long-Term ECG | 0.9764 | 0.9774 | 0.9785 |
| MIT-BIH Normal Sinus Rhythm | 0.9787 | 0.9790 | 0.9793 |
| MIT-BIH ST Change | 0.9700 | 0.9701 | 0.9701 |
| MIT-BIH Supraventricular Arrhythmia | 0.8639 | 0.8473 | 0.8448 |
| Sudden Death Patients BIH | 0.9795 | 0.9799 | 0.9804 |

It should be noted that while duplicating layers and increasing the size of a deep learning model or adding new features may slightly improve the performance of the classification task, it also increases the time required to analyze and provide results. Therefore, it is imperative to take into account the trade-off between performance and time when considering these approaches. As a matter of fact, the results obtained so far are satisfactory. However, the performance in segmenting heart conditions like supraventricular arrhythmia needs to be improved by involving more features. Besides, providing additional samples and recourse to the algorithms can lead to more precise results and interpretations. It will also help in finding solutions to achieve more accurate outcomes. At present, our analysis has relied upon F-score, precision, recall, accuracy, and computational time. Furthermore, in future endeavors, we intend to include specific computational time measurements to effectively evaluate the proposed model.

Moreover, as future work, the adopted methodology should be leveraged on alternative datasets and engaged in a collaboration with health care providers. This collaboration will be used to ascertain the

most effective means of implementing and using the methodology in real-world settings.

## VI. Conclusion

In conclusion, here we present an approach based on DL to identify and understand how the ECG signals can be classified with high accuracy and low error rate. In the current paper, the proposed methods were applied to PhysioNet's QT public database. We worked on categorizing ECG signal parameters into three main groups, namely 'PQ', 'QRST', and 'TP', and we obtained accuracy up to 96%. Future studies on various datasets and powerful computational recourses should confirm our methodology and findings.

## VII. Acknowledgment

This work was partially supported by project cyberHuman, part of the SERICS program (PE00000014) under the MUR National Recovery and Resilience Plan funded by the European Union - NextGenerationEU and by European Union through the ADAPTO project, part of the RESTART program, NextGenerationEU PNRR, CUP E63C2 2002040007, CP PE0000001.

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
