# OpenReview forum: "Frequency and Uncertainty driven Deep Learning Approach to Segment Electrocardiogram Signals for Effective Heart Parameters Estimation"
_IEEE.org/EMBS/BHI/2024/Conference — IEEE BHI'24_

### Official Review · Reviewer_Pkoq · 2024-08-07
**This paper proposes a novel deep learning approach for segmenting electrocardiogram (ECG) signals by integrating temporal, frequency, and uncertainty information. The study utilizes PhysioNet's QT database, employing multilayer deep learning architectures enhanced with short-time Fourier transform (STFT) and approximate entropy features. The results demonstrate up to 92% accuracy in classifying ECG signals, suggesting significant potential for improving heart rhythm parameter detection.**

**Overall Rating:** 8
**Confidence:** 2

**Other Quality Metrics:**

Clarity of Writing: Good
Clinical Significance: Good
Methodological Novelty: Great
Experiments and Results: Great

**Questions For The Authors:**

In the introduction section, there might be too much information that maybe it would be more fitting in the materials and method section. Please, consider to move there.

Can the authors please discuss about the computational complexity by thinking real-time application? In relation with this, is there any particular reason why the authors adopted Deep learning technicques rather than machine learning?

Feature Selection: How did you determine the optimal features (STFT and entropy) for your model? Were other features considered?

Literature review is very appreciated, but please provide a more point-to-point.  Which are the key-points compared to the existing literature? Discussing in this way could enhance the overall quality of the manuscript.

Are there any other evaluation metrics that the authors can consider for assessing their models? Can also the authors clarify on how they used 5-fold CV?

Clinical Validation: Have the authors conducted or planned any clinical validation studies to evaluate the model's performance in real-world settings? Or maybe using a different dataset with different settings (e.g. acquiring technology, settings, etc).

**Strengths:**

Innovative Approach: The integration of temporal, frequency, and uncertainty information in deep learning models for ECG segmentation is novel and promising.

High Accuracy: Achieving up to 92% classification accuracy demonstrates the effectiveness of the proposed methodologies.

Comprehensive Preprocessing: The use of noise filtering, gap removal, and normalization ensures high-quality input data for the models.

Robust Methodology: The use of multilayer architectures and extensive cross-validation enhances the reliability of the findings.

**Summary Of The Paper:**

This study aims to enhance the accuracy of ECG signal segmentation and classification by leveraging temporal, frequency, and uncertainty features. Authors employed the PhysioNet QT database, and they preprocess the ECG data through noise filtering, gap removal, and normalization. The proposed model combines convolutional and bidirectional long short-term memory (Conv-BiLSTM) networks with STFT and approximate entropy features. Extensive experimentation and five-fold cross-validation indicate that the model achieves up to 92% accuracy in classifying ECG signals into PQ, QRST, and TP segments.

**Weaknesses:**

Limited Dataset: The study relies solely on the PhysioNet QT database, which may not capture all the variability present in broader clinical settings.

Computational Complexity: The use of advanced deep learning architectures and additional features like STFT and entropy may require significant computational resources, which could limit practical applicability.

Generalization: While the model shows high accuracy, its performance on unseen or more diverse datasets remains untested.

---

### Official Review · Reviewer_WfLE · 2024-08-08
**The paper presents a novel approach for analyzing ECG signals to enhance the classification of its primary components. However, several corrections are necessary, particularly to make the results more straightforward and clearer**

**Overall Rating:** 6
**Confidence:** 4

**Other Quality Metrics:**

-	Clarity of writing. Good
-	Clinical Significance. Fair (nor part of the work)
-	Methodological Novelty. Good
-	Experiments and Results. Fair (clarifications are needed)

**Questions For The Authors:**

-	Can you add a clinical reference when describing how clinical practitioners analyze ECG to determine cardiac conditions?
-	There are missing references: 1) In the introduction, mainly in the 3rd paragraph; and 2) In the first paragraph of the II. Literature Review section.
-	It is not clear in the introduction how temporal evaluation is assessed in the work. Can you clarify this? Additionally, it would be helpful if the three dimensions (temporal, frequency, and entropy) are better introduced with the methodology followed from the beginning. This remains unclear at the end of section II.
-	Minor suggestion: Can you rephrase the last paragraph in the introduction? The methods and results are not “discussed” in that section; they are described or presented. The discussion comes in the proper section.
-	The work does not mention the quality of the dataset used. Can you provide details on this?
-	Of the records finally used, how many are annotated by the two experts?
-	Overall, the preprocessing details are okay, but the selection of the z-score normalization method is not justified. Can you indicate why this method was chosen?
-	The description of the four experiments conducted is unclear. Can you clearly describe these in chapter III?
-	Avoid including speculative sentences like “These findings could have…” in the methods section and results section. Such statements belong in the discussion section.
-	At the beginning of the Results section, “…, Several…” should not be capitalized.
-	Figures should appear after they are mentioned in the text. For example, Fig 3 is on page 4, but it is mentioned on page 5. Please correct this. Also, Fig 5 should come after Table IV.
-	The Results section is not straightforward and is mixed with some methods descriptions. Can you clarify and separate these sections better?
-	Check the grammar in the second paragraph of section V: “…, two DL models proposed for detecting ECG parameters, …”
-	The work mentions “…The present study aimed to investigate the impact of duplicating layers…” When and how is this done?

**Strengths:**

A key strength of this work is the incorporation of entropy together with the temporal and frequency analysis, which allows for a more comprehensive evaluation of the complexity of ECG signals.

**Summary Of The Paper:**

The paper introduces a robust approach to analyze ECG signals and enhance the current classification of the main signal parameters. It details the pre-processing steps and model development processes, and the model has been assessed through four different experimental settings. The proposed method appears to be accurate, although further improvements with new datasets and better computational resources are anticipated.

**Weaknesses:**

No additional work is suggested.

---

### Official Review · Reviewer_Pmcr · 2024-08-10
**Overall is good**

**Overall Rating:** 6
**Confidence:** 4

**Other Quality Metrics:**

Clarity of Writing: Great
 Clinical Significance: Good
Methodological Novelty: Good
Experiments and Results: Good

**Questions For The Authors:**

While Table 3 presents the confusion matrix for each experiment, a deeper analysis of the model's limitations or failure cases would provide a more balanced view and highlight areas for future work. Discussing these aspects can help in refining the approach and setting realistic expectations for its application.
 While the use of PhysioNet's QT database is appropriate, validation against external datasets would strengthen the generalizability of the model.
It would be highly beneficial to investigate which features play an important role in prediction. Conducting an ablation study or providing model interpretation could offer new insights and deepen the understanding of the underlying mechanisms driving the model's performance.

**Strengths:**

Achieving up to 92% accuracy in classifying ECG signals into three categories is a notable accomplishment that underscores the effectiveness of the proposed model.
The paper provides a detailed methodological description, making it reproducible and transparent. This level of detail ensures that others in the field can replicate and build upon the work.
The paper is well written and organized. It will attract border readers.

**Summary Of The Paper:**

This paper presents a deep learning approach to segment Electrocardiogram (ECG) signals for effective heart parameter estimation, effectively integrating temporal, frequency, and uncertainty data to classify ECG signals into three main categories: PQ, QRST, and TP. Utilizing PhysioNet's QT public database, the study employs preprocessing steps such as noise filtering and gap removal, and incorporates features like the Short-Time Fourier Transform and approximate entropy to enhance classification accuracy. The model achieves up to 92% accuracy, suggesting significant advancements over traditional methods.

**Weaknesses:**

Overall, this paper provides a solid contribution to the field. Here are some recommendations that could further enhance its value:
Benchmarking Against Existing Methods: The paper would benefit from expanded direct comparisons with existing state-of-the-art methods, providing clearer benchmarking against both traditional machine learning and other deep learning approaches. This could help establish the model's relative performance more definitively.
Discussion on Model Limitations: While Table 3 presents the confusion matrix for each experiment, a deeper analysis of the model's limitations or failure cases would provide a more balanced view and highlight areas for future work. Discussing these aspects can help in refining the approach and setting realistic expectations for its application.
Validation Against External Datasets: While the use of PhysioNet's QT database is appropriate, validation against external datasets would strengthen the generalizability of the model.
Feature Importance Analysis: It would be highly beneficial to investigate which features play an important role in prediction. Conducting an ablation study or providing model interpretation could offer new insights and deepen the understanding of the underlying mechanisms driving the model's performance.

---

### Decision · Program_Chairs · 2024-09-23

Accept